# The Effect of Atmospheric Carbon Dioxide Concentration on the Growth and Chlorophyll Fluorescence Characteristics of Hazelnut Leaves under Cadmium Stress

Xiaojia Liu [†], Yan Cai [†], Peiyan Ni, Binghan Liu and Xuedong Tang *

College of Horticulture, Jilin Agricultural University, Changchun 130118, China; liuxiaojia@jlau.edu.cn (X.L.); yanc@jlau.edu.cn (Y.C.)
* Correspondence: tangxd94@126.com
† These authors contributed equally to this work.

**Abstract:** To understand the response of hazelnut to the increased concentration of carbon dioxide ($CO_2$) under cadmium (Cd) pollution stress, this paper used an artificial open top chamber to control the $CO_2$ concentration (at 370 and 750 $\mu mol \cdot mol^{-1}$) and to study the effects of an elevated $CO_2$ concentration on the growth and photosynthetic capacity of hazelnut leaves under different levels of Cd stress. The results showed that the increase in atmospheric $CO_2$ concentration has a tendency to alleviate the inhibition of plant growth caused by Cd. The net photosynthetic rate rose significantly, although the transpiration rate and stomatal conductance of hazelnut leaves decreased slightly with the rise in $CO_2$ concentration. The rise in $CO_2$ concentration had no significant effect on the activity of the photosystem II (PSII) reaction center in hazelnut leaves. Under Cd stress conditions, the rise in $CO_2$ concentration significantly enhanced the PSII hazelnut leaves' photochemical activity, which promotes the PSII receptor's electron transfer capacity side and alleviates the degree of damage to the oxygen-evolving complex and the thylakoid membrane of the PSII donor side. The number of active reaction centers per unit area of hazelnut leaves, and the proportion of energy absorbed by PSII that is used for photosynthetic electron transfer, increased under severe stress conditions, which in turn reduced the energy proportion that was used for heat dissipation, providing $CO_2$'s effective fixation energy in the dark reaction. In conclusion, the rise in the $CO_2$ concentration enhances hazelnut's heavy metal resistance by improving the PSII function under Cd stress conditions.

**Keywords:** cadmium pollution; $CO_2$; hazelnut; growth; chlorophyll fluorescence

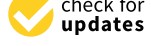



## 1. Introduction

At present, due to the combustion of fossil fuels and the change of land use types, the sedimentation rate of atmospheric carbon dioxide ($CO_2$) concentration and nitrogen (N) have reached their highest ever recorded rates and are still increasing. The atmospheric $CO_2$ concentration has increased from 315 $\mu mol/mol$ in 1959 to a current concentration of approximately 385 $\mu mol/mol$, with a growth rate of 1.9 $\mu mol/mol$ per year. The atmospheric $CO_2$ concentration is predicted to increase to 540–970 $\mu mol/mol$ by 2100 [1]. Such a high $CO_2$ concentration is bound to have a great impact on tree growth and the carbon pool of forests. Numerous $CO_2$ enrichments, nutrient addition experiments, and meta-analyses have shown that high $CO_2$ concentrations have resulted in significant fertilization effects [2–5]. Among them, a high concentration of $CO_2$ can promote plant photosynthesis, thereby improving forest productivity. However, some studies showed that despite the general increase in atmospheric $CO_2$ concentration, global tree growth and forest carbon pool did not significantly increase, suggesting that other factors, such as Cd stress and environmental pollution, may hinder the stimulation of $CO_2$ concentration on forest carbon storage in many regions [6–9]. Therefore, previous studies face limitations because (1) many studies were focused on single factors (the increase in atmospheric $CO_2$ concentration or

the impact of heavy metals), while few studies investigated dual or multiple factors, and (2) many studies have focused on forests polluted by heavy metals in tropical, subtropical, temperate, and cold regions, with little attention paid to economically important trees stressed by heavy metals. With the rapid increase in the global atmospheric $CO_2$ concentration, heavy metal pollution of the soil has also become a global environmental problem, which has caused serious damage, especially in rapidly industrialized and urbanized areas. In Northeast China, heavy metal pollution in forest ecosystems has expanded from downtown areas to suburbs and remote mountainous locations, and the contents of lead (Pb), cadmium (Cd), and zinc (Zn) far exceed critical values for plant damage and soil background [1,9,10]. Considering the fact that complex factors coexist, an increasing number of researchers believe that well-designed multi-factor composite experiments are essential to accurately predict the dynamics of vegetation carbon use [7,11–13].

Hazelnut (*Corylus* spp.) is an important economic tree species in Northeast China, with high practical value. Hazelnuts are rich in nutrients and contains high contents of fat, protein, vitamins, and mineral elements [14]. The nutrient content of hazelnuts is twice that of bread and 1.5 times that of pork. In addition, the oil content of hazelnuts is approximately 54%, which is 2–3 times that of soybean, and it is the raw material used for the production of edible oil and various industrial oils. In addition, it also has medicinal values. Unsaturated fatty acids and vitamin E, which can prevent cardiovascular and cerebrovascular diseases, delay aging, prevent vascular sclerosis, and moisturize skin, are abundant in hazelnuts. Moreover, hazelnuts contain the anti-cancer chemical taxol, which can treat and prevent cancer. The content of trace elements, such as magnesium, calcium, and potassium, in hazelnuts is very high; thus, the long-term consumption of the nut can be beneficial to blood pressure regulation. Hazelnut plays an important role in the control of deserts and the conversion of farmland to forests [14]. Hazelnut trees have developed roots and strong adaptability; therefore, they are excellent resources for wind prevention and sand fixation, prevention of water and soil loss, and conversion of farmland to forests. However, the northeast region of China has the densest population, heaviest industry, and largest urban distribution in the country, which is affected by frequent and continuous human activities [15–18]. Environmental problems, such as intensified salinization of soil, and increasingly serious heavy metal and organic matter pollution, are occurring in this region. Cd is one of the most toxic heavy metals widely existing in the environment, and is particularly toxic to plants [18–20].

Therefore, in this paper, we used open top chamber (OTC) experiments to explore the impact of an elevated $CO_2$ concentration on the growth of hazelnut under Cd pollution conditions. Our findings provide a scientific basis for vegetation restoration and management of cadmium-contaminated land.

## 2. Materials and Methods

### 2.1. Experimental Materials

The experiments were conducted in the Soil Science Experimental of Jilin Agricultural University in 2022. One-year-old hazelnut seedlings were selected and cultivated for another one year. High-quality, healthy seedlings were selected based on their strong root system (more than eight to ten lignified roots longer than 20 cm and rich in fibrous roots), vigorous stem, and plump buds, with an overall height of 80–130 cm.

**Root trimming and soaking:** Before planting, the root system of the seedlings was trimmed to 10–13 cm. The roots were soaked for 30 min before planting. The roots of the prepared seedlings were placed into a hole, allowing the root system to be spread out. The roots were covered with soil (topsoil backfill) after the seedlings were appropriately placed. The seedlings were gently firmed in by stepping on the soil. Immediately after planting, the seedlings were thoroughly irrigated, and the soil was covered with plastic film to seal the soil and preserve moisture retention to improve the survival rate of seedlings.

**Routine management of trees:** Soil loosening and timely weeding was conducted in combination with field management. During the growth period, weeding was performed

three to four times to maintain a weed-free plantation. To determine whether topdressing should be applied, the status of soil fertility was determined. Topdressing was generally not applied in the year of field planting; however, if topdressing was necessary, a small amount of nitrogen fertilizer was applied in mid-July (30–50 g per plant). Once the seedlings had survived, the plastic film was removed. Irrigation was applied when the temperature was high in late May, and drainage was carried out in a timely manner in case of waterlogging. Before freezing in winter, seedlings were irrigated once, and the roots were mounded up to prevent damage due to cold, with a mound height of about 30 cm.

**Soil management:** Tillage was conducted to keep the soil loose and free of weeds. In some cases, grass was allowed to grow between rows. If there was no intercropping of other crops between rows, rotary tillage was applied three to four times on flat land with a rotary cultivator. If grass was allowed to grow, weeding was performed three to four times with a brush cutter.

**Fertilization management:** Topdressing according to the soil fertility status was applied in the second and third years of growth. Each year, the seedlings were fertilized two to three times using the following fertilization scheme: topdressing with 200–300 g nitrogen fertilizer once or twice per plant from April to May; topdressing 150–200 g N, P, K compound fertilizer once or twice per plant from June to July. Urea was used in nitrogen fertilizer (N content 46%); Phosphorus fertilizer application contained Potassium dihydrogen phosphate ($P_2O_5$ content 52%, $K_2O$ content 34%); and potassium fertilizer application consisted of potassium sulfide ($K_2O$ content 52%).

*2.2. Experimental Design*

In this study, a near-natural method using an OTC was adopted. The OTC was composed of a base, a chamber, and an inflation system, and was cylindrical. The base was a brick-wall structure with a diameter of 3 m and a height of 0.7 m. The chamber was fixed onto the base by stainless steel poles and was wound and reinforced by wire mesh. The chamber was enclosed with transparent PVC film. The inflation system was composed of an interconnected $CO_2$ cylinder, a $CO_2$ flowmeter, PVC pipe, and a blower. A PVC pipe surrounded the chamber, and the pipe had a 0.3 cm diameter hole every 30 cm on the side facing the center of the chamber. The soil in the chamber came from the plantation and the order of soil layers (0–10, 10–30, and 30–60 cm) was maintained during the transfer process. The concentration was 370 μmol/mol, and the concentration of the $CO_2$ treatment was set to 750 μmol/mol. Since the annual sedimentation of Cd in the atmosphere is very low, Cd in the soil primarily comes from metal mines and sewage irrigation; therefore, the Cd treatment was set at 10 kg Cd/(hm$^2$·a), which is characterized as weak and Cd pollution, respectively.

The experiments included four treatments: control (CK), 10 kg Cd (Cd) treatment, Cd and $CO_2$ (Cd + $CO_2$) treatment, and CK and $CO_2$ (CK + $CO_2$) treatment. Five replicates were conducted for each treatment, yielding 20 OTCs in total. The $CO_2$ concentration in the chamber was controlled at 750 μmol/mol by the surrounding perforated rubber pipe and the $CO_2$ heating flow valve inside the growth chamber. The fumigation time was 8:00–17:00. In the control, the Cd solution was replaced by the same amount of deionized water. Seedlings were planted in August 2022, and they were experimentally treated in September 2022 after adapting and growing stably in the OTCs.

*2.3. Determination of Experimental Parameters*

**Growth parameter determination:** Fifteen plants with the same growth trend were selected for further analysis. Plant phenotypic characteristics were leaves and measurements, including leaf length and the last fully expanded leaf's width. A scanner (DS-50000, EPSON, Suwa City, Japan) was used to scan the hazelnut's leaves, and the leaf area coefficients of 0.75 were obtained through image processing. The leaf area (leaf length × leaf width × 0.75) was also calculated. The water on the surface of the plant was absorbed with filter paper, and the fresh weight was determined with an analytic balance. The samples

were subsequently placed in an oven at a temperature of 10.5 °C for a duration of 30 min in order to eliminate any moisture. Additionally, the samples were heated at 80 °C until reaching a stable weight in order to ascertain the weight when completely dried.

**Determination of photosynthetic gas exchange parameters:** Under different durations of Cd stress, the last fully expanded leaf of hazelnut seedlings was selected in each treatment. A concentration of 370 $\mu mol \cdot mol^{-1}$ $CO_2$ was fixed by $CO_2$ steel cylinders, and the light intensity (Photon Flux Density, PFD) was set to 1000 $\mu mol \cdot m^{-2} \cdot s^{-1}$ using the built-in light source of the instrument. The photosynthetic gas exchange parameters were obtained using the Li-6800 photosynthetic system (Lincoln, NE, USA). The net photosynthetic rate ($P_n$), stomatal conductance ($G_s$), transpiration rate ($T_r$), and intercellular $CO_2$ concentration ($C_i$) of hazelnut leaves in different treatments were measured, and each measurement (five separate samples) was repeated five times.

**Determination of chlorophyll fluorescence kinetics (OJIP) curve:** We chose the final fully grown leaf from hazelnut seedlings that were subjected to various treatments. Each leaf was exposed to darkness for 30 min using a dark adaptation clip. The plant efficiency analyzer (Handy PEA, Hansatech, King's Lynn, UK) was used to measure the OJIP curves of the leaves after the darkness adaptation period. The average value of the five replicates was used to plot the OJIP curve.

The relative variable fluorescence $V_J$ of point J on the $V_{O-P}$ curve (2 ms), the relative variable fluorescence $V_K$ of point K on the $V_{O-J}$ curve (0.3 ms), and the relative variable fluorescence $V_L$ of point L on the $V_{O-K}$ curve (0.15 ms) were obtained. A JIP-test analysis was conducted on the OJIP curve to obtain various parameters. These parameters are the maximum photochemical efficiency of PSII ($F_v/F_m$), performance index of PSII based on absorption ($PI_{ABS}$), performance index of electron flux to the final PSI electron acceptors, i.e., of both PSII and PSI ($PI_{total}$), and the number of active response centers per unit area ($RC/CS_m$). JIP-test analysis of OJIP curves was performed using the method described by Strasser et al. [6].

*2.4. Data Analysis*

Statistical analyses were conducted using Excel 2007 and SPSS 23.0 software. Two-way analysis of variance (ANOVA) was employed to compare the variations among different data sets.

## 3. Results

*3.1. Growth Characteristics*

The $CO_2$ concentration had little impact on the growth characteristics of hazelnut leaves during the experimental period. In the high $CO_2$ concentration treatment, the dry weight of hazelnut leaves increased, but not significantly. Compared with the CK, the fresh weight and dry weight of hazelnut leaves under Cd stress significantly decreased by 21.7% and 19.3%, respectively (Figure 1).

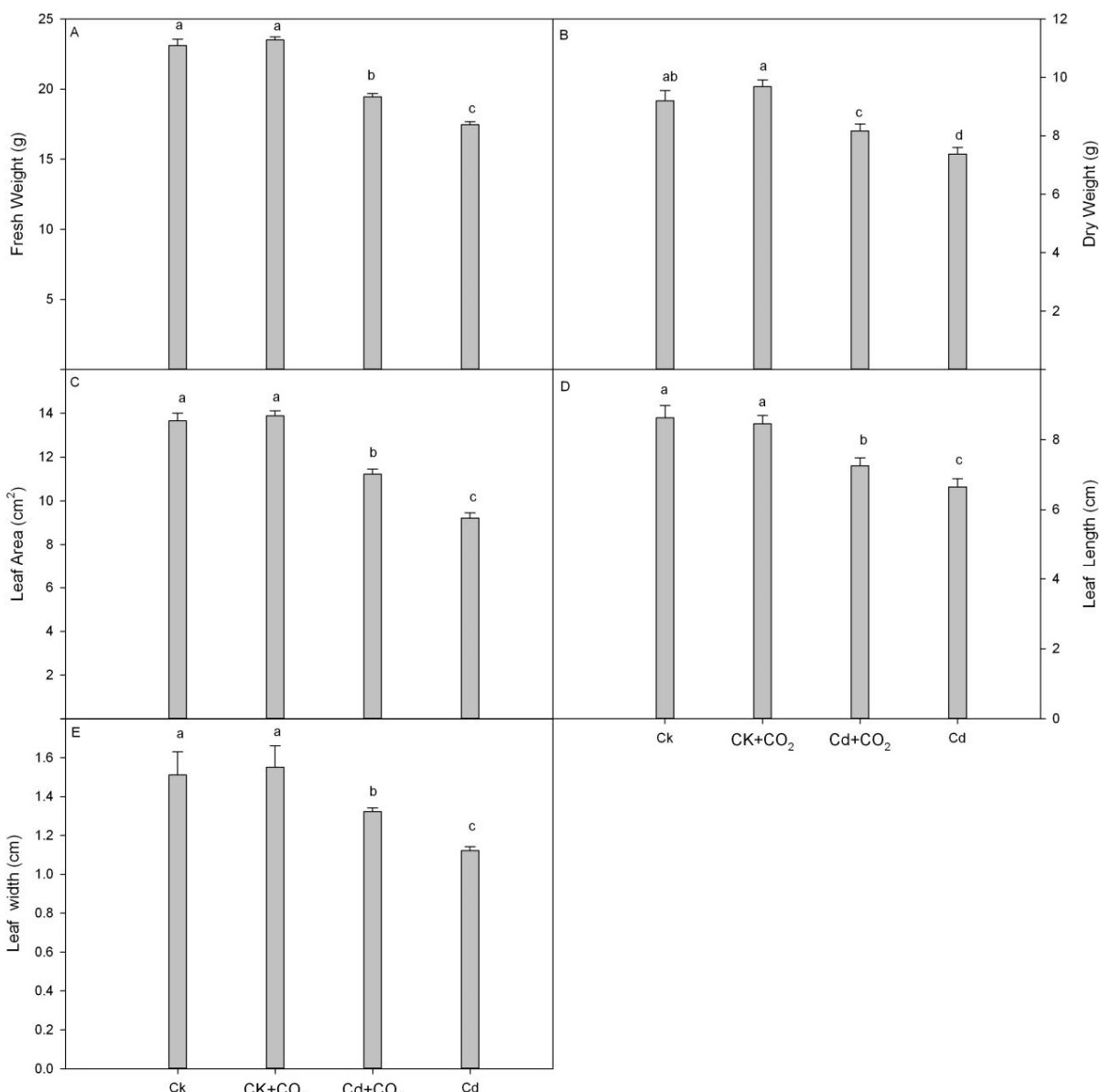

**Figure 1.** The interaction of high $CO_2$ concentration and Cd stress with the growth characteristics of hazelnut leaves. (**A**): Fresh weight; (**B**): Dry weight; (**C**): Leaf area; (**D**): Leaf length; (**E**): Leaf width. CK + $CO_2$: CK Under 750 μmol·mol$^{-1}$ $CO_2$ concentrations; Cd + $CO_2$: Cd stress + 750 μmol·mol$^{-1}$ $CO_2$ concentrations; Cd: Cd stress. Note: Data in the figure are the mean ± SE; values followed by different lowercase letters indicate a significant difference ($p < 0.05$).

### 3.2. Photosynthetic Gas Exchange Parameters

The hazelnut leaves subjected to a $CO_2$ concentration of 750 μmol·mol$^{-1}$ had slightly higher rates of photosynthesis (Pn) and internal $CO_2$ concentration ($C_i$) compared with those subjected to a $CO_2$ concentration of 370 μmol·mol$^{-1}$, with a significant increase in $P_n$ by 9.41% ($p < 0.05$). Conversely, when exposed to cadmium (Cd) stress, the Pn, stomatal conductance ($C_i$), transpiration rate ($T_r$), and stomatal conductance ($G_s$) of hazelnut leaves all exhibited a significant decrease. However, hazelnut leaves treated with a $CO_2$ concentration of 750 μmol·mol$^{-1}$ displayed different responses compared with those treated with a $CO_2$ concentration of 370 μmol·mol$^{-1}$ (Figure 2).

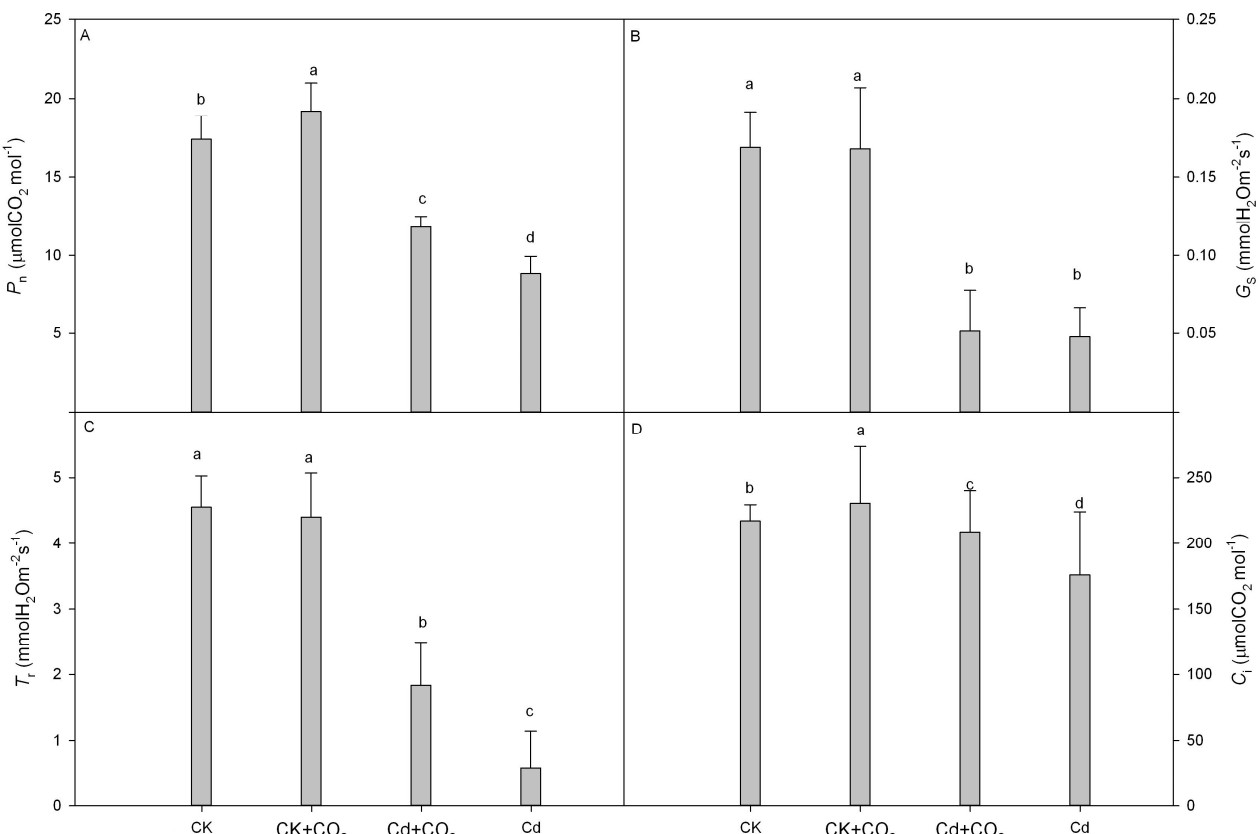

**Figure 2.** The interaction of high $CO_2$ concentration and Cd stress with the photosynthetic gas exchange parameters of hazelnut leaves. (**A**): Net photosynthetic rate ($P_n$); (**B**): Stomatal conductance ($G_s$); (**C**): Transpiration rate ($T_r$); (**D**): Intercellular $CO_2$ concentration ($C_i$). Note: Data in the figure are the mean ± SE; values followed by different lowercase letters indicate a significant difference ($p < 0.05$).

*3.3. Chlorophyll Fluorescence Characteristics*

The analysis showed that when exposed to a $CO_2$ concentration of 370 μmol·mol$^{-1}$ and Cd stress, the $F_o$ of hazelnut leaves increased by 16.12% (significantly different at $p < 0.05$) compared with the control group, while $F_m$ decreased by 17.88% (significantly different at $p < 0.05$). The change in $F_o$ and $F_m$ was less pronounced in hazelnut leaves under the 750 μmol·mol$^{-1}$ $CO_2$ treatment compared with the 370 μmol·mol$^{-1}$ $CO_2$ treatment (Figure 3).

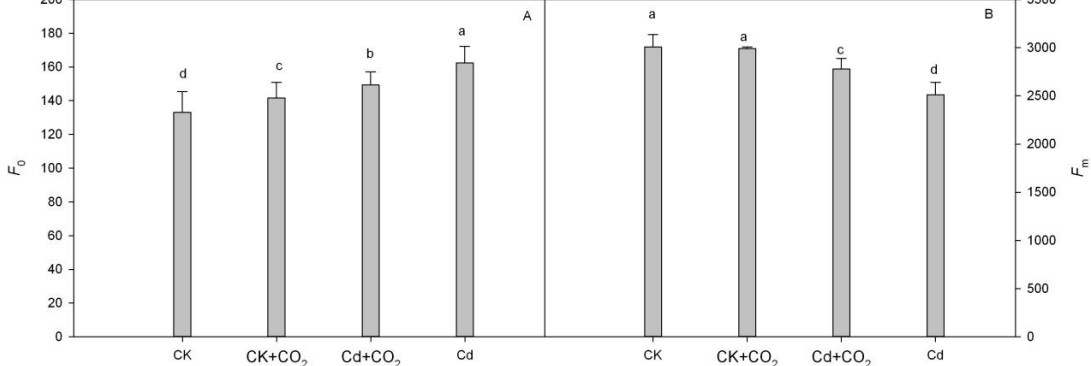

**Figure 3.** The interaction of high $CO_2$ concentration and Cd stress with the characteristic points (O and P). Parameters of hazelnut leaves. (**A**): Minimal recorded fluorescence intensity; (**B**): Maximal recorded fluorescence intensity. CK: Under 370 μmol·mol$^{-1}$ $CO_2$ concentrations. Note: Data in the figure are the mean ± SE; values followed by different lowercase letters indicate a significant difference ($p < 0.05$).

### 3.4. PSII Photochemical Efficiency

There was no significant difference in the $F_v/F_m$, $PI_{ABS}$, and $PI_{total}$ of hazelnut leaves when comparing the CK treatment to the 370 and 750 $\mu mol \cdot mol^{-1}$ $CO_2$ treatments. However, as the duration of the Cd stress treatment increased, there was a noticeable decrease in these parameters in the hazelnut leaves. When exposed to Cd stress, the hazelnut leaves treated with 750 $\mu mol \cdot mol^{-1}$ $CO_2$ showed a 5.78%, 25.13%, and 5.61% decrease in $F_v/F_m$, $PI_{ABS}$, and $PI_{total}$, respectively, compared with those treated with 370 $\mu mol \cdot mol^{-1}$ $CO_2$, indicating a significantly larger difference in these parameters (Figure 4).

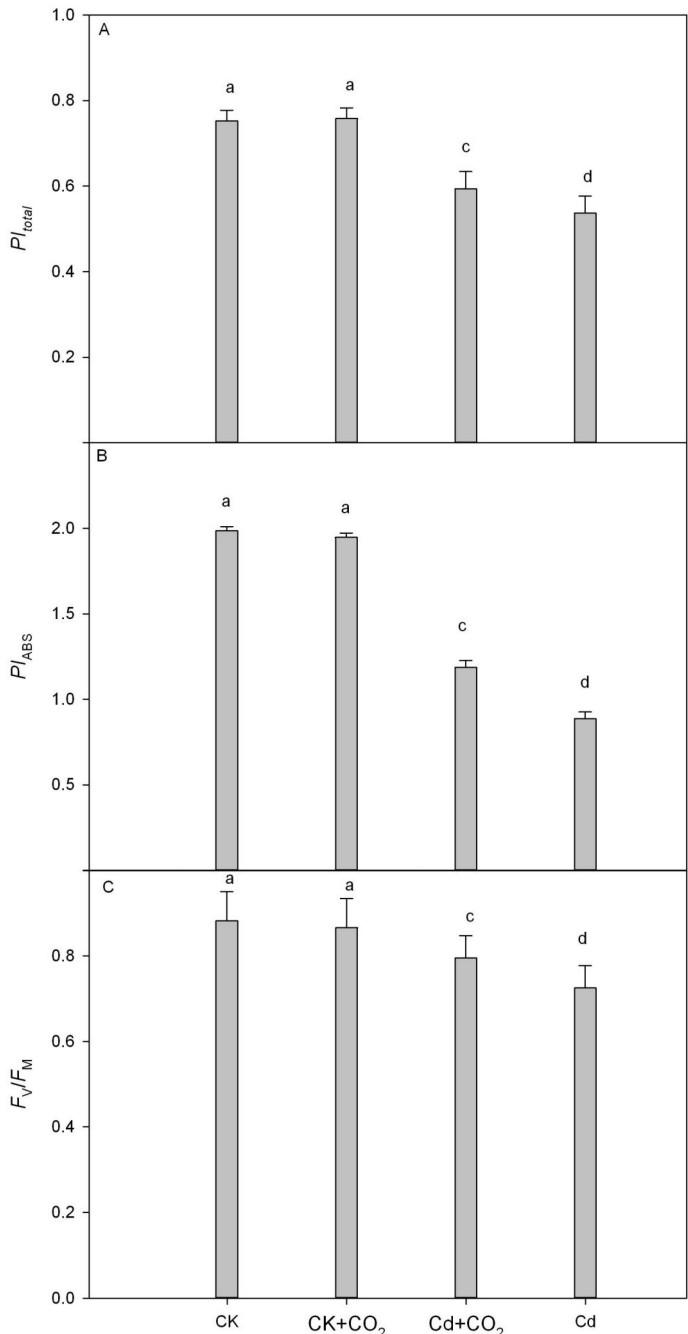

**Figure 4.** The interaction of high $CO_2$ concentration and Cd stress with PSII photochemical efficiency. Parameters of hazelnut leaves. $PI_{total}$: electron flux to the final PSI electron acceptors, i.e., of both PSII and PSI (**A**); $PI_{ABS}$: PSII based on absorption (**B**); $F_v/F_m$: the maximum photochemical efficiency of PSII (**C**). Note: Data in the figure are the mean $\pm$ SE; values followed by different lowercase letters indicate a significant difference ($p < 0.05$).

### 3.5. Standardized O-P Curve and Relative Variable Fluorescence $V_J$, $V_K$, and $V_L$

Analysis of the quantitative changes in $V_J$ revealed that the $V_J$ of hazelnut leaves significantly increased under Cd stress conditions in both the 370 and 750 $\mu mol \cdot mol^{-1}$ $CO_2$ treatments compared with the control group (CK). However, the increase in $V_J$ was greater in the 370 $\mu mol \cdot mol^{-1}$ $CO_2$ treatment than in the 750 $\mu mol \cdot mol^{-1}$ $CO_2$ treatment. Analysis of the quantitative changes in $V_K$ indicated that the $V_K$ of hazelnut leaves in the 750 $\mu mol \cdot mol^{-1}$ $CO_2$ treatment was 8.45% lower ($p < 0.05$) than that in the 370 $mol \cdot mol^{-1}$ $CO_2$ treatment under Cd stress conditions.

Quantitative analysis of the changes in $V_L$ showed that the $V_L$ of hazelnut leaves increased significantly by 5.31% and 7.23% ($p < 0.05$) in the 370 $\mu mol \cdot mol^{-1}$ $CO_2$ treatment under Cd stress conditions compared with the control group. On the other hand, the $V_L$ of hazelnut leaves did not show a significant change in either the Cd stress or Cd stress conditions in the 750 $\mu mol \cdot mol^{-1}$ $CO_2$ treatment (Figure 5).

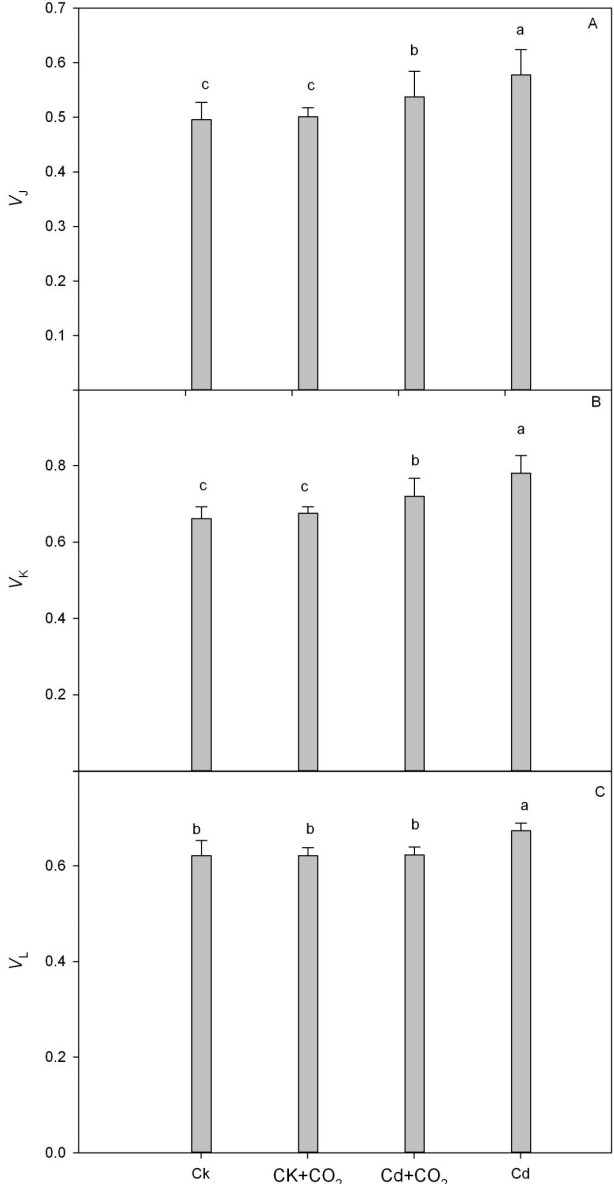

**Figure 5.** The interaction of high $CO_2$ concentration and Cd stress with relative variable fluorescence $V_J$ (**A**), relative variable fluorescence $V_K$ (**B**), and variable fluorescence $V_L$ (**C**) of hazelnut leaves. Note: Data in the figure are the mean $\pm$ SE; values followed by different lowercase letters indicate a significant difference ($p < 0.05$).

### 3.6. Parameters of Energy Distribution per Unit of Reaction Center

The change in $RC/CS_m$ suggested that under Cd stress, the $RC/CS_m$ of hazelnut leaves under the 370 μmol·mol$^{-1}$ $CO_2$ treatments decreased by 17.98% compared with the CK ($p < 0.05$). No significant decrease in the $RC/CS_m$ was observed in the 750 μmol·mol$^{-1}$ $CO_2$ treatment (Figure 6).

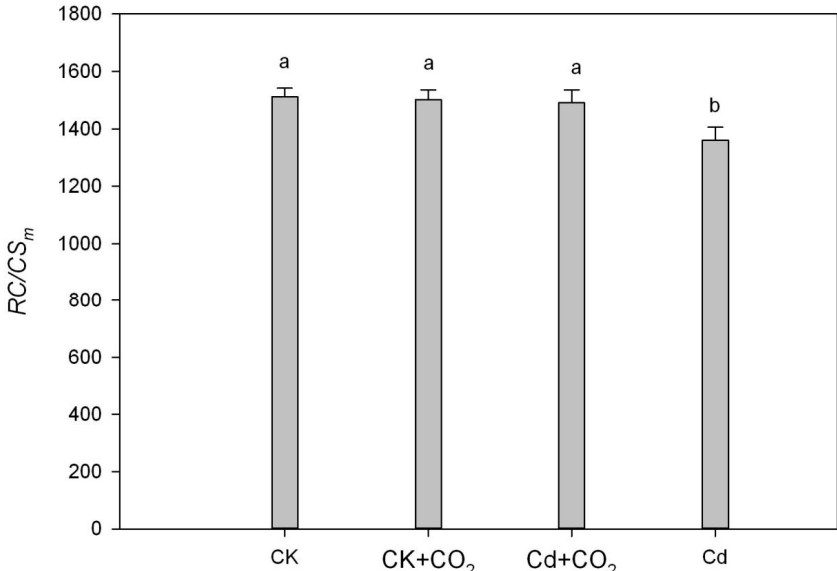

**Figure 6.** Effects of high $CO_2$ concentration, Cd stress, and their interaction on energy distribution per unit of reaction center and per unit area of hazelnut leaves. Note: Data in the figure are the mean ± SE; values followed by different lowercase letters indicate a significant difference ($p < 0.05$).

## 4. Discussion

The results of this study showed the biomass accumulation and growth of hazelnut that grew in Cd-contaminated soils to have different responses to the rise in atmospheric $CO_2$ concentration. The rise in biomass of hazelnut that was treated with Cd was lower than that of the control, indicating that Cd stress conditions have an inhibitory effect on hazelnut's growth. The migration of heavy metal Cd in the soil–plant system affects the biochemical and physiological processes, growth, and development of plants directly by impeding root growth, inhibiting nutrient absorption and water, and inhibiting photosynthesis, thereby reducing plant yield [21–23]. These results suggested that in Cd-contaminated soil, a high concentration of $CO_2$ can alleviate the toxic effect of Cd on plants.

### 4.1. The Growth and Development of Hazelnut under Cd Stress Conditions

The rise in atmospheric $CO_2$ concentration and Cd content in the soil is the trend change of our future climate. Numerous studies have shown that Cd stress has a significant inhibitory effect on plant development and growth. For instance, some studies found that with water content's continuous reduction, plant height, root/shoot ratio, and aboveground and belowground fresh weight all showed a downward trend in *Coreopsis tinctoria* [24]. The biomass and morphological indexes of seedlings decreased significantly under Cd stress. A study on phenotypes of *Zea mays* L.' seedlings found that Cd stress greatly inhibited the development and growth of wheat seedlings, which was mainly reflected by the number of tillers, seedling height, and leaf area. Other reports suggested a fertilizer effect of the increased $CO_2$ concentration on crop growth [25]. The present study found that under high $CO_2$ concentrations, the leaf length, leaf width, and leaf area of hazelnut significantly rose, indicating that the rise in photosynthetic area is conducive to dry matter's accumulation in the plant, thus increasing the fresh weight and dry weight of the aboveground part of hazelnut.

*4.2. Effects of High $CO_2$ Concentration on Gas Exchange Parameters under Cd Stress*

A few studies have shown that a rise in $CO_2$ concentration causes a decrease in the partial closure and the $G_s$ of stomata. Some studies showed that the doubling of $CO_2$ concentration reduced the stomatal conductance by an average of 11% and the rise in $CO_2$ concentration reduced winter wheat's stomatal density [26,27]. In this study, in the 750 μmol·mol$^{-1}$ $CO_2$ treatment, $G_s$ and $T_r$ of hazelnut leaves were unchanged compared with those in the 370 μmol·mol$^{-1}$ $CO_2$ treatment; however, $P_n$ was significantly increased. This indicated that $G_s$ might not be the limiting factor for the rise in $P_n$ under the double $CO_2$ concentration treatment. In addition, an elevated $CO_2$ concentration can improve the Rubisco activity, is related to carboxylation, and can enhance photosynthetic capacity by inhibiting photorespiration and increasing substrates. $CO_2$ is both the substrate for photosynthesis and stomata's regulator. Some studies showed that the effects of Cd stress and elevated $CO_2$ concentration significantly interact, impacting $P_n$, and that a high $CO_2$ concentration improves plants' adaptability to Cd stress [28].

*4.3. Effects of High $CO_2$ Concentration on PSII Photochemical Efficiency under Cd Stress Conditions*

The presence of Cd stress inhibited the dark reaction, resulting in a decrease in activity in the PSII reaction center. The accumulation of assimilates (NADPH and ATP) also had a feedback effect, further inhibiting the light reaction process [29–31]. In this experiment, as the duration of Cd stress increased, hazelnut leaves exhibited a decreasing trend in $F_v/F_m$, $PI_{ABS}$, and $PI_{total}$. Particularly, the decrease in $PI_{ABS}$ and $PI_{total}$ was more pronounced, indicating a decrease in photochemical activity in the PSII of hazelnut leaves under Cd stress. A previous study similarly found that an increased $CO_2$ concentration affects the photosynthetic energy conversion and electron transfer of plant leaves. However, in this experiment, when the Cd stress was mild, the rise in $CO_2$ concentration had no significant effect on $F_v/F_m$, $PI_{ABS}$, and $PI_{total}$ of hazelnut leaves. Under Cd stress, hazelnut leaves treated with 750 μmol·mol$^{-1}$ $CO_2$ had significantly higher values for $F_v/F_m$, $PI_{ABS}$, and $PI_{total}$ compared with those treated with 370 μmol·mol$^{-1}$ $CO_2$. This suggests that the increased $CO_2$ concentration can alleviate photoinhibition in hazelnut leaves under Cd stress.

Under stress conditions, the blocked sites of photosynthetic electron transfer in plants often appear on the electron donor side and the electron acceptor side of the PSII reaction center [32,33]. As the specified symbol, the oxygen-evolving complex (OEC) on the PSII electron donor side was injured, and the rise of the relative variable fluorescence at point K of 0.3 ms ($V_K$) was considered. The relative variable fluorescence at the OJIP curve's 2 ms, point J ($V_J$), represented the degree of the active reaction center's closure, and the rise of $V_J$ indicated that the electron transfer from $Q_A$ to $Q_B$ in the photosynthetic electron transfer chain was inhibited and that the accumulation of redox-state $Q_A$ gradually increased [32,33]. In this experiment, $V_J$ and $V_K$ in hazelnut leaves rose to varying degrees under Cd stress, indicating that the decrease in PSII photochemical activity in hazelnut leaves caused by Cd stress was related to blockage of electron transfer on the PSII donor and receptor sides. Although $V_J$ and $V_K$ of hazelnut leaves showed no significant difference in the 370 and 750 μmol·mol$^{-1}$ $CO_2$ treatments, under Cd stress, the $V_K$ and $V_J$ of leaves in the 750 μmol·mol$^{-1}$ $CO_2$ treatment were both significantly lower than in the 370 μmol·mol$^{-1}$ $CO_2$ treatment. These results showed that the rise in $CO_2$ concentration could alleviate the degree of damage to the OEC in hazelnut leaves under Cd stress, and that it could promote the electron transfer from $Q_A$ to $Q_B$ on the PSII receptor side under Cd stress.

The rise of the relative variable fluorescence at point L ($V_L$) was thought of as the change in thylakoid membrane fluidity, which was the major indicator that its functional and structural integrity was destroyed [32,33]. In this experiment, it was observed that the $V_L$ of hazelnut leaves decreased significantly under Cd stress under the treatment with a $CO_2$ concentration of 750 μmol·mol$^{-1}$ compared with the treatment with a $CO_2$ concentration of 370 μmol·mol$^{-1}$. The increase in $V_L$ was considered an important indication of the change in fluidity of the thylakoid membrane, as well as the damage to its functional and

structural integrity. Therefore, the increase in $CO_2$ concentration was found to enhance the stability of the thylakoid membrane in hazelnut leaves under Cd stress. The stability of the thylakoid membrane is directly linked to the stability of PSII function. Consequently, the relatively stable thylakoid membrane also contributes to the stability of PSII.

The utilization and absorption of light energy in plant leaves' PSII reaction center can be affected by stress conditions [34–36]. Under Cd stress conditions, the hazelnut leaves' $RC/CS_m$ significantly decreased, indicating that when a part of the reaction centers in hazelnut leaves becomes inactive under Cd stress, the leaves could adapt to Cd stress by enhancing the light energy absorption capacity using the remaining active reaction centers. These results are similar to several previously published studies. Compared with the results from the 370 $\mu mol \cdot mol^{-1}$ $CO_2$ treatment, related and overall similar results were obtained with regard to the trend of light energy absorption and allocation parameters of the PSII reaction centers in hazelnut leaves under the 750 $\mu mol \cdot mol^{-1}$ $CO_2$ treatment, but the magnitude of variation was significantly reduced. In particular, the $RC/CS_m$ was not significantly reduced in the $CO_2$ treatments combined with Cd stress. One possible explanation for this outcome is the hypothesis that the increase in $CO_2$ levels may impact the way hazelnut leaves absorb and distribute light energy through the photosystem II (PSII) reaction centers when under cadmium stress. This could lead to a decrease in the number of reaction centers affected by this stress. As a result of this process, the amount of energy absorbed by the PSII reaction centers for electron transfer is also expected to increase. This ensures a regular energy supply and the production of assimilatory power (in the form of ATP and NADPH) during PSII electron transfer, which aids in the process of $CO_2$ fixation [37,38].

### 5. Conclusions

The increase in $CO_2$ concentration enhances the photosynthetic performance of hazelnut leaves. However, in the absence of Cd stress, the increase in $CO_2$ concentration has a minimal effect on the photochemical activity and electron transfer of hazelnut leaves' PSII. On the other hand, when subjected to Cd stress and elevated $CO_2$ concentrations, the photosynthetic electron transfer on the PSII receptor side is stimulated, while the damage to the OEC on the PSII donor side is alleviated. As a result, the energy distribution of the PSII reaction centers is optimized, leading to an increase in the quantity of active reaction centers and the stability of the thylakoid membrane.

**Author Contributions:** Conceptualization, X.T.; Methodology, X.L. and P.N.; Formal analysis, X.L. and B.L.; Investigation, Y.C.; Resources, X.T.; Writing—original draft, X.L. and Y.C.; Writing—review and editing, X.T. and X.L. All authors have read and agreed to the published version of the manuscript.

**Funding:** This research was funded by Jilin Provincial Natural Science Foundation grant number No.YDZJ202101ZYTS113.

**Institutional Review Board Statement:** Not applicable.

**Data Availability Statement:** The data that support the findings of this study are available from the corresponding author upon reasonable request.

**Conflicts of Interest:** The authors declare no conflict of interest.

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
