# Peer review of "The Effect of Atmospheric Carbon Dioxide Concentration on the Growth and Chlorophyll Fluorescence Characteristics of Hazelnut Leaves under Cadmium Stress"

_forests, doi:10.3390/f14091791_

Round 1

Reviewer 1 Report

The manuscript "Effect of atmospheric carbon dioxide concentration on the 2 growth and chlorophyll fluorescence characteristics of hazelnut 3 leaves under cadmium stress"  of Xiaojia et al., dscribed the changes in hazelnut under Cd effect, CO2 stress and its combination.

Some recomendation listed below:

1. It will be better to use two-way ANOVA or MANOVA test for presentation the difference between stressors on parameters.

2. There are not the data about Cd concentration in leaves. According the methods the Cd spread because it is difficult to identified the amount of Cd that absorbed on the leaves. Moreover the wax layer on the leaves also prevent to metals absobtion. At the same time will be interesting to understand how changed the Cd concentration in leaves under CO2 stress? Perhaps in this case the concentration can be less than under Cd effect alone?

Also, maybe the possitive effect of CO2+Cd on some parameters can be a result of lower Cd concentration?

Because the detection of Cd concentraton in leaves can help to better discription of results.

3. If it is possible, it is better to added informatin about effect of combination CO2 + different heavy metals, not only Cd, on parameters that were presented in this manuscript.

4. What is potential mechanism due to CO2 in combination with Cd, alleviate its toxicity?

Author Response

Dear Editor:

Thank you very much for reviewing our manuscript Manuscript. Those comments are all valuable and very helpful for revising and improving our paper, as well as the important guiding significance to our researches. We have studied comments carefully and have made correction which we hope meet with approval. Revised portion are marked in yellow in the paper. The main corrections in the paper and the responds to the reviewer’s comments are as flowing:

Reviewers' comments:Comments to the Author 

Reviewer: 1
The manuscript "Effect of atmospheric carbon dioxide concentration on the 2 growth and chlorophyll fluorescence characteristics of hazelnut 3 leaves under cadmium stress"  of Xiaojia et al., dscribed the changes in hazelnut under Cd effect, CO2 stress and its combination. Some recomendation listed below:

1.It will be better to use two-way ANOVA or MANOVA test for presentation the difference between stressors on parameters.

Answer: According to the Suggestions given by the reviewer, we made a recount calculation of all the data using the two-way ANOVA statistical analysis method. And the experimental results are reanalyzed.We reanalyzed the data and revised the discussion section accordingly.

Line 191-196;204-209;301-314;317-319;325-329;336-347;352-354;363-380;382-389

2.There are not the data about Cd concentration in leaves. According the methods the Cd spread because it is difficult to identified the amount of Cd that absorbed on the leaves. Moreover the wax layer on the leaves also prevent to metals absobtion. At the same time will be interesting to understand how changed the Cd concentration in leaves under CO2 stress? Perhaps in this case the concentration can be less than under Cd effect alone?

Also, maybe the possitive effect of CO2+Cd on some parameters can be a result of lower Cd concentration?

Because the detection of Cd concentraton in leaves can help to better discription of results.

Answer: Thank you very much for your opinion, which will be very helpful to our future experimental design. We will adopt your suggestion in the next experiment, and maybe we will do a detailed experiment separately according to your suggestion.

3.If it is possible, it is better to added informatin about effect of combination CO2 + different heavy metals, not only Cd, on parameters that were presented in this manuscript.

Answer: Thank you very much for your valuable advice. In the design of the experiment, we considered the experimental treatment of heavy metal pollution of different concentrations such as Cd and Pb, and tried to analyze the effects of different kinds of heavy metals and different concentrations on plant photosynthetic organs, as well as the corresponding regulation modes of plants.

4.What is potential mechanism due to CO2 in combination with Cd, alleviate its toxicity?

Answer: Our research concluded that elevated CO2 concentrations and Cd stress promoted the photosynthetic electron transfer on the PSII receptor side and  alleviated the damage to the OEC on PSII donor side. Finally,  the PSII reaction centers' energy distribution was optimized, and active reaction centers' quantity and the stability of thylakoid membrane rose.In conclusion, the increase in the CO2 concentration enhances the heavy metal resistance of hazelnut by improving the PSII function under Cd stress conditions.

We tried our best to improve the manuscript and made some changes in the manuscript. These changes will not influence the content and framework of the paper. And here we did not list the changes but marked in yellow in revised paper.

We appreciate for Editors and Reviewers’ warm work earnestly, and hope that the correction will meet with approval.

Once again, thank you very much for your comments and suggestions.

Yours sincerely

Xuedong Tang

Reviewer 2 Report

The investigation and findings reported in the MS ( Effect of atmospheric carbon dioxide concentration on the growth and chlorophyll fluorescence characteristics of hazelnut leaves under cadmium stress) assume significance climate change and  heavy metal toxicity (stress). The has been well crafted. However, I find following constraints before accepting MS for publication:

Clarify whether the dose of nitrogen fertilizer denotes N content or fertilizer amount (line 103, 115, 141). Also mention the name of nitrogen fertilizer. Also mention individual amount of N, P K in the fertilizer dose and source name (line 116).

Provide real time photographs of the experimental lay out and Open Top CO2 (OTC) as supplementary material with appropriate mention in the MS text.

 Cadmium compound (is it CdCl2 as mentioned on line 140?) used for treatment and whether 5 and 10 kg doses refer to cadmium or cadmium compound (line 132-133, 142).

Is leaf area calculation mentioned on line 148-149) standardize by the authors or taken from the published literature? In either case provide adequate reproducible information. Check calculation of leaf area values given Fig. 1.

The authors are advised to revise Materials and Method section addressing above queries.

Author Response

Dear Editor:

Thank you very much for reviewing our manuscript Manuscript. Those comments are all valuable and very helpful for revising and improving our paper, as well as the important guiding significance to our researches. We have studied comments carefully and have made correction which we hope meet with approval. Revised portion are marked in yellow in the paper. The main corrections in the paper and the responds to the reviewer’s comments are as flowing:

Reviewers' comments:Comments to the Author 

Reviewer: 2 Comments to the Author 
The investigation and findings reported in the MS ( Effect of atmospheric carbon dioxide concentration on the growth and chlorophyll fluorescence characteristics of hazelnut leaves under cadmium stress) assume significance climate change and  heavy metal toxicity (stress). The has been well crafted. However, I find following constraints before accepting MS for publication:

1.Clarify whether the dose of nitrogen fertilizer denotes N content or fertilizer amount (line 103, 115, 141). Also mention the name of nitrogen fertilizer. Also mention individual amount of N, P K in the fertilizer dose and source name (line 116).

Answer: According to the requirements of the reviewer, the mistake has been corrected in the revised MS.

“Since the annual sedimentation of Cd in the atmosphere is very low, Cd in the soil primarily comes from metal mines and sewage irrigation; therefore, the Cd treatment 10 kg Cd/(hm2·a), which characterized weak and Cd pollution, respectively.

The experiments included four treatments: control (CK), 10 kg Cd (Cd) treatment, Cd and CO2 (Cd+CO2) treatment, and CK and CO2 (CK+CO2) treatment. Five replicates were conducted for each treatment, yielding 20 OTCs in total. Three vigorous one-year-old hazelnut seedlings were selected, evenly spaced, and planted in each OTC. The CO2 concentration in the chamber was controlled to 750 μmol/mol by the surrounding perforated rubber pipe and CO2 heating flow valve inside the growth chamber. The fumigation time was 8:00–17:00. In the control, the Cd solution was replaced by the same amount of deionized water. Seedlings were planted in Auguest 2022, and they were experimentally treated in September 2022 after adapting and growing stably in the OTCs.”

2.Provide real time photographs of the experimental lay out and Open Top CO2 (OTC) as supplementary material with appropriate mention in the MS text.

Answer: Thank you very much for your suggestion, we will provide the corresponding experimental OTC photos.

FIG. 1OTC gas chamber was used before, and later the experimental design was improved, as shown in FIG. 2. For the experimental design, an external air conditioner can be added to adjust the temperature. The carbon dioxide cylinder is released by a circular PVC pipe, and the PVC pipe will drill a round hole with a diameter of 3 mm every 10 cm.   

3.Cadmium compound (is it CdCl2 as mentioned on line 140?) used for treatment and whether 5 and 10 kg doses refer to cadmium or cadmium compound (line 132-133, 142).

Answer: According to the requirements of the reviewer, the mistake has been corrected in the revised MS.

“The CO2 in the gas cylinder was sent into the PVC pipe using a blower, and the CO2 evenly diffused into the growth chamber through small holes. The soil in the chamber came from the plantation and the order of soil layers (0–10, 10–30, and 30–60 cm) was maintained during the transfer process. The concentration was 370 μmol/mol, and the concentration of the CO2 treatment was set to 750 μmol/mol. Since the annual sedimentation of Cd in the atmosphere is very low, Cd in the soil primarily comes from metal mines and sewage irrigation; therefore, the Cd treatment 10 kg Cd/(hm2·a), which characterized weak and Cd pollution, respectively.”

4.Is leaf area calculation mentioned on line 148-149) standardize by the authors or taken from the published literature? In either case provide adequate reproducible information. Check calculation of leaf area values given Fig. 1..The authors are advised to revise Materials and Method section addressing above queries.

Answer: According to the requirements of the reviewer, the mistake has been corrected in the revised MS.

“15 plants with the same growth trend were selected for further analysis. Plant phenotypic characters were measured, including leaf length and leaf width of the last fully expanded leaf; A scanner (DS-50000, EPSON, Japan) was used to scan the leaves of hazelnut, and the leaf area coefficients of 0.75 were obtained through image processing. the leaf area (leaf length × leaf width × 0.75) was also calculated.”

We tried our best to improve the manuscript and made some changes in the manuscript. These changes will not influence the content and framework of the paper. And here we did not list the changes but marked in yellow in revised paper.

We appreciate for Editors and Reviewers’ warm work earnestly, and hope that the correction will meet with approval.

Once again, thank you very much for your comments and suggestions.

Yours sincerely

Xuedong Tang

Reviewer 3 Report

The manuscript deals with an important and interesting topic and is relatively well-written. Still, some problems must be addressed before acceptance for publication.

1. The biological significance of some chlorophyll fluorescence-related measurements should be explained more explicitly - especially of those that are shown in Figure 3 and Figure 5. What exactly do these measurements tell about the functioning of the photosystem? 

2. The reference in line 177 - "by Strasser (1995)" - should be rather given as "by Strasser et al. [6]", in order to find it more easily in the reference list.

3. There are some unclear details in Materials and Methods. For example:

a) in line 132, it is stated "the Cd treatment was set at 5 kg and 10 kg...", although the only experimental Cd treatment, as described further, was 10 kg";

b) in line 146, it is stated "25 plants with the same growth trend were selected for further analysis". How many plants were from each treatment? It could not be 25, because, as described in lines 134-136, each treatment was represented by five replicates (ATCs) with three plants in each OTC. Thus, according to this decription, threre were 15 plants for each treatment. What was then represented by 25 plants mentioned in line 146?

c) in lines 159 and 164-165, it is not stated clearly whether the measurement repetitions were biological (five separate samples) or just anaytical;

d) it is not described or cited, how photosynthetic gas exchange parameters (PnGs, Tr) were measured (lines 153-159). Besides, the meaning of "under different durations of Cd stress" in lines 153-154 is not clear. What were these different durations?

4. The descriptions of results often seem not corresponding well with the data given in Figures. For example, in lines 185-187, it is stated "High CO2 concentration significantly increased leaf length, leaf width, and leaf area...". None of these can be seen in Figure 1, at least, not under cadmium-free treatment. This tendency to describe the differences that are statistically insignificant is seen through the whole Results section, and also in Discussion and in Conclusions (e.g., the statement "the rise in CO2 concentration reduced hazelnut leaves' stomatal conductance..." (line 386) is not supported by the data in Figure 2). The authors should check this carefully, and re-describe the results in such a manner that a description would be clearly supported by the data in the figures. 

5. Some statements in Discussion require clarification or restatement. Examples:

a) it is stated in lines 296-297, "In this study, the root growth of hazelnut was significantly inhibited with the increased duration of Cd stress". However, neither  in Methods, nor in Results this study claimed to have measured Cd effect on hazelnut root growth.

b) the sentence in line 326 should be better started with "Other studies indicate that..." or similar wording, if the described findings (in lines 326-328) come not from this study but from [31-33]. Still, I also cannot see how, for example, the reference No. 33 (which is about buffalo nutrition) is directly related to what is described here.

The general quality of English is good; still, the text should be checked thoroughly and some sentences (as, e.g., in lines 305-308 and in lines 352-355) rewritten.

Author Response

Dear Editor:

Thank you very much for reviewing our manuscript Manuscript. Those comments are all valuable and very helpful for revising and improving our paper, as well as the important guiding significance to our researches. We have studied comments carefully and have made correction which we hope meet with approval. Revised portion are marked in yellow in the paper. The main corrections in the paper and the responds to the reviewer’s comments are as flowing:

Reviewers' comments:Comments to the Author 

Reviewer: 3
Comments to the Author 
The manuscript deals with an important and interesting topic and is relatively well-written. Still, some problems must be addressed before acceptance for publication.

1.The biological significance of some chlorophyll fluorescence-related measurements should be explained more explicitly - especially of those that are shown in Figure 3 and Figure 5. What exactly do these measurements tell about the functioning of the photosystem? 

Answer: According to the requirements of the reviewer, we have made modified in the article.

“The relative variable fluorescence VJ of point J on the VO-P curve (2 ms), the relative variable fluorescence VK of point K on the VO-J curve (0.3 ms), and the relative variable fluorescence VL of point L on the VO-K curve (0.15 ms) were obtained. ”

“The increase of the relative variable fluorescence at point K of 0.3 ms (VK) was thought of as the specified symbol that the oxygen-evolving complex (OEC) on PSII electron donor side was injured. The relative variable fluorescence at 2 ms of the OJIP curve, point J (VJ) represented the degree of closure of the active reaction center and the increase of VJ indicated that the electron transfer from QA to QB in the photosynthetic electron transfer chain was inhibited and the accumulation of redox-state QA gradually increased[34,35].”

“The increase of relative variable fluorescence of point L (VL) was thought of as the change of thylakoid membrane fluidity, which was the major indicator that its structural and functional integrity was destroyed[34,35].

2.The reference in line 177 - "by Strasser (1995)" - should be rather given as "by Strasser et al. [6]", in order to find it more easily in the reference list.”

Answer: According to the requirements of the reviewer, we have made modified in the article.

“and the number of the active response centers per unit area (RC/CSm). JIP-test analysis of OJIP curves was performed using the method described by Strasser et al.[6].”

  1. 3.There are some unclear details in Materials and Methods. For example:
  2. a) in line 132, it is stated "the Cd treatment was set at 5 kg and 10 kg...", although the only experimental Cd treatment, as described further, was 10 kg";

Answer: According to the requirements of the reviewer, the mistake has been corrected in the revised MS.

“Since the annual sedimentation of Cd in the atmosphere is very low, Cd in the soil primarily comes from metal mines and sewage irrigation; therefore, the Cd treatment 10 kg Cd/(hm2·a), which characterized weak and Cd pollution, respectively.”

  1. b) in line 146, it is stated "25 plants with the same growth trend were selected for further analysis". How many plants were from each treatment? It could not be 25, because, as described in lines 134-136, each treatment was represented by five replicates (ATCs) with three plants in each OTC. Thus, according to this decription, threre were 15 plants for each treatment. What was then represented by 25 plants mentioned in line 146?

Answer: According to the requirements of the reviewer, the mistake has been corrected in the revised MS.

“15 plants with the same growth trend were selected for further analysis. Plant phenotypic characters were measured, including leaf length and leaf width of the last fully expanded leaf”

  1. c) in lines 159 and 164-165, it is not stated clearly whether the measurement repetitions were biological (five separate samples) or just anaytical;

Answer: According to the requirements of the reviewer, the mistake has been corrected in the revised MS.

“using the built-in light source of the instrument. the photosynthetic gas exchange parameters by Li-6800 photosynthetic system (Licor Corporation, UK). The net photosynthetic rate (Pn), stomatal conductance (Gs), transpiration rate (Tr), and intercellular CO2 concentration (Ci) of blueberry leaves in different treatments were measured, Each measurement (five separate samples) was repeated five times.”

  1. d) it is not described or cited, how photosynthetic gas exchange parameters (Pn, Gs, Tr) were measured (lines 153-159). Besides, the meaning of "under different durations of Cd stress" in lines 153-154 is not clear. What were these different durations?

Answer: According to the requirements of the reviewer, the mistake has been corrected in the revised MS.

“using the built-in light source of the instrument. the photosynthetic gas exchange parameters by Li-6800 photosynthetic system (Licor Corporation, UK). The net photosynthetic rate (Pn), stomatal conductance (Gs), transpiration rate (Tr), and intercellular CO2 concentration (Ci) of blueberry leaves in different treatments were measured, Each measurement (five separate samples) was repeated five times.”

4.The descriptions of results often seem not corresponding well with the data given in Figures. For example, in lines 185-187, it is stated "High CO2 concentration significantly increased leaf length, leaf width, and leaf area...". None of these can be seen in Figure 1, at least, not under cadmium-free treatment. This tendency to describe the differences that are statistically insignificant is seen through the whole Results section, and also in Discussion and in Conclusions (e.g., the statement "the rise in CO2 concentration reduced hazelnut leaves' stomatal conductance..." (line 386) is not supported by the data in Figure 2). The authors should check this carefully, and re-describe the results in such a manner that a description would be clearly supported by the data in the figures. 

AnswerAnswer: According to the requirements of the reviewer, the mistake has been corrected in the revised MS.

“The CO2 concentration had little impact on the growth characteristics of hazelnut leaves during the one-month experimental period. In the high CO2 concentration treatment, the dry weight of hazelnut leaves increased significantly. Compared with the CK, the fresh weight and dry weight of hazelnut leaves under Cd stress significantly decreased by 21.7% and 19.3%, respectively. Under Cd stress, the increase in CO2 concentration could alleviate the inhibition of Cd on the character of hazelnut leaves.”

5.Some statements in Discussion require clarification or restatement. Examples:

  1. a) it is stated in lines 296-297, "In this study, the root growth of hazelnut was significantly inhibited with the increased duration of Cd stress". However, neither  in Methods, nor in Results this study claimed to have measured Cd effect on hazelnut root growth.

AnswerAnswer: According to the requirements of the reviewer, the mistake has been corrected in the revised MS.We reanalyzed the data and revised the discussion section accordingly

“A few studies have shown that an rise in CO2 concentration causes a decrease in the partial closure and the Gs of stomata. Some studies showed that the doubling of CO2 concentration reduced the stomatal conductance by an average of 11% and the rise in CO2 concentration reduced winter wheat's stomatal density [29]. In this study, in the 750 μmol·mol-1 CO2 treatment, Gs and Tr of hazelnut leaves were unchanged than that in the 370 μmol·mol-1 CO2 treatment, however, with a significantly increased Pn. This indicated that Gs might not be the limiting factor for the rise in Pn under the double CO2 concentration treatment. Besides, elevated CO2 concentration can improve the Rubisco activity and is related to carboxylation, and can enhance photosynthetic capacity by inhibiting photorespiration and increasing substrates. CO2 is both the substrate for photosynthesis and are stomata's regulator. Some studies showed that the effects of Cd stress and elevated CO2 concentration significantly interacts, impacting Pn, and high CO2 concentration improves plants' adaptability to Cd stress [30].” 

  1. b) the sentence in line 326 should be better started with "Other studies indicate that..." or similar wording, if the described findings (in lines 326-328) come not from this study but from [31-33]. Still, I also cannot see how, for example, the reference No. 33 (which is about buffalo nutrition) is directly related to what is described here.

The general quality of English is good; still, the text should be checked thoroughly and some sentences (as, e.g., in lines 305-308 and in lines 352-355) rewritten.

Answer: According to the requirements of the reviewer, the mistake has been corrected in the revised MS.

“Other studies indicate that the dark reaction was inhibited under Cd stress, and the accumulation of assimilates (NADPH and ATP) would feedback and inhibit the light reaction process,” 

“A few studies have shown that an rise in CO2  concentration causes a decrease in the partial closure and the Gs of stomata. Some studies showed that the doubling of CO2  concentration reduced the stomatal conductance by an average of 11% and the rise in CO2  concentration reduced winter wheat's stomatal density [29]. In this study, in the 750 μmol·mol-1 CO2  treatment, Gs and Tr of hazelnut leaves were unchanged than that in the 370 μmol·mol-1 CO2  treatment, however, with a significantly increased Pn. This indicated that Gs might not be the limiting factor for the rise in Pn under the double CO2 concentration treatment. Besides, elevated CO2  concentration can improve the Rubisco activity and is related to carboxylation, and can enhance photosynthetic capacity by inhibiting photorespiration and increasing substrates. CO2 is both the substrate for photosynthesis and are stomata's regulator. Some studies showed that the effects of Cd stress and elevated CO2 concentration significantly interacts, impacting Pn, and high CO2 concentration improves plants' adaptability to Cd stress [30].” 

We reanalyzed the data and revised the discussion section accordingly.

Line 191-196;204-209;301-314;317-319;325-329;336-347;352-354;363-380;382-38

We tried our best to improve the manuscript and made some changes in the manuscript. These changes will not influence the content and framework of the paper. And here we did not list the changes but marked in yellow in revised paper.

We appreciate for Editors and Reviewers’ warm work earnestly, and hope that the correction will meet with approval.

Once again, thank you very much for your comments and suggestions.

Yours sincerely

Xuedong Tang

Round 2

Reviewer 1 Report

All recommendations were accepted. I wish further to make an experiment with metal detection and publish new manuscript.

Author Response

Dear Editor:

Thank you very much for reviewing our manuscript Manuscript. Those comments are all valuable and very helpful for revising and improving our paper, as well as the important guiding significance to our researches. We have studied comments carefully and have made correction which we hope meet with approval. Revised portion are marked in yellow in the paper. The main corrections in the paper and the responds to the reviewer’s comments are as flowing:

Reviewers' comments:Comments to the Author 

Reviewer: 1 

All recommendations were accepted. I wish further to make an experiment with metal detection and publish new manuscript.

Answer: Thank you again for your valuable feedback. We will conduct subsequent experiments based on your suggestions and submit a new manuscript as soon as possible.

Reviewer: 2

I evaluated the revised MS. The authors did a good job revising the MS. However, the revision has not addressed my query "Clarify whether the dose of nitrogen fertilizer denotes N content or fertilizer amount (line 103, 115, 141). Also mention the name of nitrogen fertilizer. Also mention individual amount of N, P K in the fertilizer dose and source name (line 116)". In addition, the lines 159-160 of the revised MS mention "blue berry" which is beyond my comprehension. Both queries need to be addressed and a calibrated response is required.

Answer: According to the requirements of the reviewer, the mistake has been corrected in the revised MS.

“Urea used in nitrogen fertilizer (N content 46%); Phosphorus fertilizer application Potassium dihydrogen phosphate (P2O5 content 52%, K2O content 34%); potassium Fertilizer application of potassium sulfide (K2O content 52%).” 

We tried our best to improve the manuscript and made some changes in the manuscript. These changes will not influence the content and framework of the paper. And here we did not list the changes but marked in yellow in revised paper.

We appreciate for Editors and Reviewers’ warm work earnestly, and hope that the correction will meet with approval.

Once again, thank you very much for your comments and suggestions.

Yours sincerely

Xuedong Tang

Reviewer 2 Report

I evaluated the revised MS. The authors did a good job revising the MS. However, the revision has not addressed my query "Clarify whether the dose of nitrogen fertilizer denotes N content or fertilizer amount (line 103, 115, 141). Also mention the name of nitrogen fertilizer. Also mention individual amount of N, P K in the fertilizer dose and source name (line 116)". In addition, the lines 159-160 of the revised MS mention "blue berry" which is beyond my comprehension. Both queries need to be addressed and a calibrated response is required.

Author Response

(The authors gave the same response as above.)

Reviewer 3 Report

The authors made most of the suggested corrections, but some work of text editing still must be done. For example:

1. The last part of the sentence in lines 132-133 is grammatically incorrect, even meaningless.

2. Regarding the number of plants used for growth parameter determination, "25" was changed to "15". Still, if 15 was the number of the whole lot of plants in each experimental treatment, then what is the meaning of the words "were selected" (line 144)?

3. The sentence in lines 157-158 is incomplete (lacking any verb).

4. If experiments were made with hazelnuts, why is it written in lines 159-160 that "blue-berry leaves in different treatments were measured"?

5. In Results, the sentence in lines 188-189 does not seem to be supported by the data in Figure 1B (the difference between control and higher CO2 concentration is not significant here).

6. In line 192, the expression "could alleviate" is not the best choice to describe the actually obtained results.

7. The sentence in lines 199-201 seems problematic, both because of its hardly correct grammatical structure and because the difference between the the control  and increased CO2 concentration in Figure 2A visually does not seem to equal 29.41 %.

8. From the sentence in lines 203-204, it is not clear what "decline is relatively small".

9. In Discussion, the sentence in lines 299-301 is grammatically incorrect.

10. The two sentences in lines 305-308 are both grammatically incorrect.

11. The sentence in lines 311-313 remains unclear. How could it describe findings from references No. 31-33, if none of these three articles dealt with Cd stress?

12. In Conclusions, it is unclear why the authors claim that "the rise in CO2 concentration reduced hazelnut leaves' stomatal conductance". This claim is not supported by the data in  Figure 2B.

13. The phrase "hazelnut's electron transfer leaves" (lines 376-377) does not seem to be grammatically correct.

14. Finally, I strongly recommend to include explanations for parameter abbreviations in the Figures. For example, in the present variant, a reader has to look elsewhere in the text to check what is the meaning of abbreviations "Pn", "Gs", "Tr" and "Ci" in Figure 2. It would be much better to have these abbreviations explained in the captions of the Figures themselves.

Language-related notes and suggestions are given in the section above.

Author Response

Dear Editor:

Thank you very much for reviewing our manuscript Manuscript. Those comments are all valuable and very helpful for revising and improving our paper, as well as the important guiding significance to our researches. We have studied comments carefully and have made correction which we hope meet with approval. Revised portion are marked in yellow in the paper. The main corrections in the paper and the responds to the reviewer’s comments are as flowing:

Reviewers' comments:Comments to the Author 

Reviewer: 1 

All recommendations were accepted. I wish further to make an experiment with metal detection and publish new manuscript.

Answer: Thank you again for your valuable feedback. We will conduct subsequent experiments based on your suggestions and submit a new manuscript as soon as possible.

Reviewer: 2

I evaluated the revised MS. The authors did a good job revising the MS. However, the revision has not addressed my query "Clarify whether the dose of nitrogen fertilizer denotes N content or fertilizer amount (line 103, 115, 141). Also mention the name of nitrogen fertilizer. Also mention individual amount of N, P K in the fertilizer dose and source name (line 116)". In addition, the lines 159-160 of the revised MS mention "blue berry" which is beyond my comprehension. Both queries need to be addressed and a calibrated response is required.

Answer: According to the requirements of the reviewer, the mistake has been corrected in the revised MS.

“Urea used in nitrogen fertilizer (N content 46%); Phosphorus fertilizer application Potassium dihydrogen phosphate (P2O5 content 52%, K2O content 34%); potassium Fertilizer application of potassium sulfide (K2O content 52%).” 

Reviewer: 3

The authors made most of the suggested corrections, but some work of text editing still must be done. For example:

  1. The last part of the sentence in lines 132-133 is grammatically incorrect, even meaningless.

Answer: According to the requirements of the reviewer, the mistake has been corrected in the revised MS.

  1. Regarding the number of plants used for growth parameter determination, "25" was changed to "15". Still, if 15 was the number of the whole lot of plants in each experimental treatment, then what is the meaning of the words "were selected" (line 144)?

Answer: According to the requirements of the reviewer, the mistake has been corrected in the revised MS.

  1. The sentence in lines 157-158 is incomplete (lacking any verb).

Answer: According to the requirements of the reviewer, the mistake has been corrected in the revised MS.

  1. If experiments were made with hazelnuts, why is it written in lines 159-160 that "blue-berry leaves in different treatments were measured"?

Answer: According to the requirements of the reviewer, the mistake has been corrected in the revised MS.

  1. In Results, the sentence in lines 188-189 does not seem to be supported by the data in Figure 1B (the difference between control and higher CO2 concentration is not significant here).

Answer: According to the requirements of the reviewer, the mistake has been corrected in the revised MS.

  1. In line 192, the expression "could alleviate" is not the best choice to describe the actually obtained results.

Answer: According to the requirements of the reviewer, the mistake has been corrected in the revised MS.

  1. The sentence in lines 199-201 seems problematic, both because of its hardly correct grammatical structure and because the difference between the the control  and increased CO2 concentration in Figure 2A visually does not seem to equal 29.41 %.

Answer: According to the requirements of the reviewer, the mistake has been corrected in the revised MS. Entry errors due to our negligence

  1. From the sentence in lines 203-204, it is not clear what "decline is relatively small".

Answer: According to the requirements of the reviewer, the mistake has been corrected in the revised MS.

  1. In Discussion, the sentence in lines 299-301 is grammatically incorrect.

Answer: According to the requirements of the reviewer, the mistake has been corrected in the revised MS.

  1. The two sentences in lines 305-308 are both grammatically incorrect.

Answer: According to the requirements of the reviewer, the mistake has been corrected in the revised MS.

  1. The sentence in lines 311-313 remains unclear. How could it describe findings from references No. 31-33, if none of these three articles dealt with Cd stress?

Answer: According to the requirements of the reviewer, the mistake has been corrected in the revised MS. We re-reviewed the references and made correct citations.

  1. In Conclusions, it is unclear why the authors claim that "the rise in CO2 concentration reduced hazelnut leaves' stomatal conductance". This claim is not supported by the data in  Figure 2B.

Answer: Thank you very much for your valuable comments! According to the requirements of the reviewer, the mistake has been corrected in the revised MS.

  1. The phrase "hazelnut's electron transfer leaves" (lines 376-377) does not seem to be grammatically correct.

Answer: According to the requirements of the reviewer, the mistake has been corrected in the revised MS.

  1. Finally, I strongly recommend to include explanations for parameter abbreviations in the Figures. For example, in the present variant, a reader has to look elsewhere in the text to check what is the meaning of abbreviations "Pn", "Gs", "Tr" and "Ci" in Figure 2. It would be much better to have these abbreviations explained in the captions of the Figures themselves.

Answer:Thank you very much for your valuable comments, which we have introduced in the second part of the article, and based on your comments we have added explanations of the corresponding abbreviations in the corresponding legend leaves in the article.

We tried our best to improve the manuscript and made some changes in the manuscript. These changes will not influence the content and framework of the paper. And here we did not list the changes but marked in yellow in revised paper.

In addition, we carried out an article duplication check, removing references our duplication result was 27% (including our own published articles), with no more than 5% duplication in a single article. The report has been uploaded via an attachment.

We appreciate for Editors and Reviewers’ warm work earnestly, and hope that the correction will meet with approval.

Once again, thank you very much for your comments and suggestions.

Yours sincerely

Xuedong Tang
